# Multi-Material Metamaterial Topology Optimization to Minimize the Compliance and the Constraint of Weight: Application of Non-Pneumatic Tire Additive-Manufactured with PLA/TPU Polymers

**DOI:** 10.3390/polym15081927

**Published:** 2023-04-18

**Authors:** Shokouh Dezianian, Mohammad Azadi

**Affiliations:** Faculty of Mechanical Engineering, Semnan University, Semnan 35131-19111, Iran

**Keywords:** multi-material, metamaterial, topology optimization, non-pneumatic tire, 3D printing

## Abstract

In non-pneumatic tires, metamaterial cells could replace the pneumatic part of the tire. In this research, to achieve a metamaterial cell suitable for a non-pneumatic tire with the objective function of increasing compressive strength and bending fatigue lifetime, an optimization was carried out for three types of geometries: a square plane, a rectangular plane, and the entire circumference of the tire, as well as three types of materials: polylactic acid (PLA), thermoplastic polyurethane (TPU), and void. The topology optimization was implemented by the MATLAB code in 2D mode. Finally, to check the quality of cell 3D printing and how the cells were connected, the optimal cell fabricated by the fused deposition modeling (FDM) method was evaluated using field-emission scanning electron microscopy (FE-SEM). The results showed that in the optimization of the square plane, the sample with the minimum remaining weight constraint equal to 40% was selected as the optimal case, while in the optimization of the rectangular plane and the entire circumference of tire, the sample with the minimum remaining weight constraint equal to 60% was selected as the optimal case. From checking the quality of 3D printing of multi-materials, it was concluded that the PLA and TPU materials were completely connected.

## 1. Introduction

As an engineering issue, optimization is basically divided into three topics: size optimization, shape optimization, and topology optimization. In size optimization, the variables are the dimensions of the structure such as thickness, surface, and volume. Shape optimization determines the shape of the structure boundaries. Topology optimization in discrete structures examines the number and optimal connection of members [1].

Researchers have recently realized that by taking inspiration from nature, cellular structures can be designed in such a way that they have novel characteristics [1]. In addition, with optimization methods, all types of cells with desired properties can be designed. So far, many cells have been discovered and designed, and are generally divided into four categories: foams, lattice, Triply Periodic Minimal Surfaces (TPMS) planes, and TPMS skeletons [2,3].

Metamaterials arise from the replication of cells next to each other [1,4]. Owing to the complex structure of metamaterials, 3D printers are used to produce them. For example, a type of metamaterial that twists under pressure has been produced with a 3D printer [5].

Among the applications of metamaterials is in car tires. Metamaterial cells replace the pneumatic part of the tire and create non-pneumatic tires. One of the advantages of these types of tire is eliminating the worry of a flat tire and the need to adjust the air pressure inside the tire. These tires are notable for the easy method of manufacturing [6]. In addition, non-pneumatic tires are more environmentally friendly due to lower raw material usage in their production [7,8]. Bras and Cobert [9] investigated the environmental impact of the Tweel tire during its life cycle, through the possibility of making these tires with recyclable and reusable materials. They concluded that non-pneumatic tires help to reduce the serious problems of the environmental pollution caused by the indestructibility of traditional tires [9]. Furthermore, the discussion of the environmental impact perfectly matches with the new sustainable management and treatment actions for end-of-life tires, such as the end-of-waste directives in Europe and the USA [10] or the waste disposal and public cleansing law in Japan [11].

These tires also have other advantages such as reducing rolling resistance and thus reducing fuel consumption and emissions [6,7,12,13,14].

There have been many activities to investigate non-pneumatic tires. One of the tires that has been designed is the Tweel tire. This tire has been examined from different aspects. This includes the possibility of manufacturing them with 3D printers [15], checking the mechanical behavior by a dynamic finite element model [16], checking the three-point bending properties and simulating this test using the finite element method [17], the effects of cell thickness on vertical stiffness and maximum local stress in cells and their weight [18], the noise produced by Tweel tires [19], investigation into different parameters affecting rolling resistance [20] and prediction of pressure distribution in the cell of this tire [21].

The Tweel tire was also designed for use on the moon, which led to a new structure called TweelTM. This tire has also been examined from different aspects, including modeling and simulating the dynamic interaction between tire and sand [22], the interaction of the tire shear layer with sand [23], examining different patterns to find a cell with a high shear bending force bearing considering the strain energy distribution [24] and investigating the ribbed metamaterial shear layer and selecting the shear layer with a specific cell geometry with the best pressure distribution in the contact area [25].

The use of optimization methods for the design of metamaterial cells has also been investigated. Cells have been optimized for one or more materials. Tavakli [26] developed a multi-material optimization algorithm. Numerical results show that the proposed algorithm is very close to the discrete scheme or 0–1. Zhengtong et al. [27] proposed an optimization algorithm for a truss foundation. In this method, the optimization of the use of different materials is considered. Chen et al. [28] presented a multi-material topology optimization algorithm for cell design by reducing stress concentration. This metamaterial cell also had the property of a negative Poisson ratio.

Chung and Du [29] developed a topology optimization framework for the design of multi-material cellular structures. The resulting cellular structure is a thermal metamaterial that must be exposed to temperature changes. Mansouri et al. [30] compressed a multi-material cellular structure. The results show that the multi-material cellular structure had better compressive properties. These multi-material structures had flexibility and load-bearing characteristics. Gao et al. [31] proposed a topology fatigue optimization approach. In this method, stress, volume, and fatigue strength are considered limitations and minimum compliance is the objective function.

In other research, Gao et al. [32] proposed a design method for the design of three-material auxetic metamaterials using the isogeometric topology optimization (ITO) method. Huang and Li [33] developed an algorithm for designing new multi-material topology optimization. Li et al. [34] also investigated the optimization of the topology of auxetic cellular composites using the surface set method.

Nguyen et al. [35] presented an approach for unit cell design using isogeometric analysis and parametric surface adjustment. Vogiatzis et al. [36] proposed a surface set-based method for topology optimization of single-material and multi-material metamaterials with a negative Poisson ratio (NPR). Zhang et al. [37] presented a new bi-material cell design method for chiral auxetic metamaterials. Zheng et al. [38] systematically investigated several isotropic materials concerning the elastic modulus and Poisson ratio. In this research, a method of topology optimization and optimal design of cells was used.

According to the literature, the studies for the optimization of metamaterial cells were carried out with:different optimization methods [24,29,30,31,32,33,34],different materials (one to several materials) [23,26,27,36],different objective functions and constraints [25,28].

Consequently, the innovation of the current research is related to:the optimization of the metamaterial cell with three materials and its possible application in non-pneumatic tires,the production and the examination of how to connect materials and the quality of 3D printing.

## 2. Materials and Methods

### 2.1. Samples

The main goal of this research is to design a non-pneumatic tire with an optimization method consisting of several materials. Different forces are applied to the tire. However, in this research, the compressive force caused by the weight of the car and the passengers and the bending force caused by the friction between the tire and the road were considered. In this article, it was assumed that the tire was moving along a straight road. As a result, lateral force was omitted. In future articles the impact of lateral force could also be investigated. According to this hypothesis and a review of articles, the dimensions and amount of force on the tire were selected [12,13,15,16,18,21,23,39,40,41,42,43,44,45,46,47,48,49,50,51,52,53,54,55,56,57,58]. These forces and the dimensions of the tire in question are shown in Figure 1. The combined compressive and bending force was selected as 1 kN. Considering that the problem was two-dimensional (2D), a concentrated load was applied.

To optimize the non-pneumatic tire, three types of solution methods were considered. In the first case (Strategy 1), a non-pneumatic tire was obtained by optimizing a square plane and putting together the optimized response in the pneumatic part of the tire (after optimization, this cell can be used in a scaled form in the tire). In the second case (Strategy 2), a rectangular plane was chosen equal to the height of the pneumatic tire section, which was obtained by putting together the optimized response in the radial direction of the non-pneumatic tire. In the third mode (Strategy 3), optimization for the entire circumference of the tire was included.

### 2.2. Materials Specifications

In this research, to optimize and find a suitable cell to replace the pneumatic part of the tire, three types of materials including polylactic acid (PLA), thermoplastic polyurethane (TPU) and void were used. The producer of both types of polymeric filaments was the YS company. PLA is among the harder 3D printable polymers on the market. The Poisson ratio for this material is 0.36, its density is 1240 kg/m^3^ [59], and its melting temperature is between 180 and 230 °C [60]. TPU polymers show properties such as toughness, strength, and wear resistance. The Poisson ratio of these materials is 0.3897, their melting temperature is 200–220 °C, and their density is 1200 kg/m^3^ [61].

In the present article, the tensile properties of the materials, which have a more critical effect, were used. Tensile test samples were prepared according to the ISO-527 standard at a speed of 50 mm/min and fused deposition modeling (FDM) method with a repeatability of three tests with 3D printing parameters according to Table 1. In the 3D printer used, the extruder is moved by the belt system in the X-Y axes and the bed is moved by the lead screw in the Z axis. This printer is able to print parts with dimensions of 550 × 600 × 700 mm. It has an accuracy of 50 and 40 μm in the X-Y and Z axes, respectively. This device was designed and manufactured by the research laboratory of Advanced Materials Behavior (AMB) at the Faculty of Mechanical Engineering, Semnan University, Semnan, Iran. The results of the tensile test are shown in Table 2. These properties were used in the optimization.

### 2.3. Optimization Procedures

Metamaterials can be designed by using optimization processes. In these processes, optimization was performed for one cell, then by repeating and putting this cell together, a metamaterial structure was fabricated. Topology optimization was performed based on finite element analysis and sensitivity analysis. In each step, the properties of the elements were specified. Mathematical programming techniques were also used to obtain convergence of the response. More information is provided in Appendix A [62,63,64,65,66,67,68].

To perform optimization in the MATLAB software, the code written by Zuo and Saitou [69] was used. In this code, it is possible to optimize several materials in 2D mode. Therefore, the optimization of the three geometries in question was defined as shown in Figure 2. The parameters used in the optimization are also presented in Table 3.

One solution to prevent rasterization of the response is to use filters such as the neighborhood radius filter. In this filter, the color or density of each element is calculated as an average of the density of adjacent elements. The neighborhood radius is one of the optimization parameters. If there is a limitation in the construction method, a smaller neighborhood radius can be used. Moreover, if there is no limit to production, the neighborhood radius can be used more. In this case, the answer is more accurate. The neighborhood radius filter considers constructability in the optimization process. The larger the radius is, the more elements are involved. As a result, more elements have the same properties. If a smaller radius is used, the boundary between the void and the solid material is clearer. However, if this radius is too small, the elements will not be grouped. Figure 3 is presented for a better understanding of this method. For the optimization, the neighborhood radius filter with the symbol of R was used.

The code used contains two types of constraints: volume and price. However, in the present study, only the volume was used. To eliminate the effect of the price constraint, a value of 50% was considered. This means that both PLA and TPU materials should be used equally. The final answer should include 50% of the cheaper material and naturally 50% of the more expensive material. In addition, the normal vector related to the price of materials was defined as (1 1 1). This means that the price of all three: void, PLA and TPU, was assumed to be the same.

To optimize the square plane, a 1 × 1 m plane was selected, which was subjected to a compressive and bending force of 1 kN. By extruding the optimized 2D geometry, the desired cell was calculated. This cell is to be used in a scaled form in the tire. Similarly, for the optimization of the rectangular plane, a plane with dimensions of 4.5 × 60 mm was selected, and the desired geometry was obtained from the rotation of this plane around the axis of symmetry (see Figure 1). This plane was also subjected to a compressive and bending force of 1 kN.

Optimizing the entire circumference of a tire with dimensions of 650 × 650 mm was considered, and the desired result was obtained by extruding its optimized response to the width of the tire section. In the design of the tire, continuous contact between the tire and the road is very important for the safety of the car and its passengers. Therefore, a barrier was considered at the point of contact between the tire and the road to prevent the separation of these two. In addition, another restriction was also considered in the center of the tire to prevent the tire from sliding on the road.

In the MATLAB code used to investigate the effect of the length of the element on the optimization response, various analyses were performed, the results of which are shown in Figure 4. According to this figure, an element size of 25 mm was chosen.

### 2.4. Printing Quality

To check the quality of 3D printing in the production of parts made of PLA and TPU and to check how to connect them, the optimal cell fabricated by the FDM method was observed using field emission scanning electron microscopy (FE-SEM) (Zeiss Sigma 300 HV, Jena, Germany). A gold coating was applied on the surface of the samples prior to SEM analysis.

## 3. Results and Discussion

### 3.1. Outputs for Square Plane

A square plane with dimensions of 1 × 1 m with an element length of 25 mm, which is equivalent to the number of 40 elements in each direction, was used for optimization with different volumes. Figure 5 illustrates the optimization results for various weight constraints. According to this figure, the sample with less remaining weight equal to 20% and 60% had discontinuity. This discontinuity requires support according to the manufacturing method (FDM), as in metamaterials it is very difficult, if not impossible, to separate these types of support from the main part due to the repetition of the cell and the complexity of the part. One of the ways to overcome this challenge is to use the constraint related to additive manufacturing. In this type of constraint, the overhang angle (the angle between the material and the horizontal line) is defined. In this case, angles greater than 45° that require support will not be fabricated along with the specimen. Therefore, the sample with a remaining weight of less than 40% was selected as the optimal sample.

This problem was implemented in a completely similar way in the ABAQUS software for both 2D and 3D modes. The problem defined in the ABAQUS software is shown in Figure 6. The answer obtained by the MATLAB code compared to the optimized cell with ABAQUS software is depicted in Figure 7.

According to Figure 7, where the quality of the optimization results is compared, the results are similar to each other to an acceptable extent. Therefore, the MATLAB code was used in 2D mode to optimize PLA and TPU materials along with voids in the target plane. Since the sample with 40% of the original weight was selected as the optimal sample in the optimization of the square plane, the same constraint was used in this step as well. As shown in Figure 7a, the weight of the sample in 3D mode (ABAQUS) was 486 g. while the weight of the solid sample was 1240 g. Therefore, it can be concluded that the remaining weight was about 40% of the solid weight specimen. The convergence diagram of the objective function is shown in Figure 8 and the result obtained is also depicted in Figure 9. According to the results, the answer obtained can only be implemented with the help of 3D printers with two nozzles.

The answer for the optimization of a square plate consisting of PLA and TPU materials was fabricated in two scales by the FDM 3D printer. The print parameters were considered according to Table 1. These cells are demonstrated in Figure 10.

### 3.2. Outputs for Rectangular Plane

In optimization with MATLAB code, a rectangular plane with dimensions of 4.5 × 60 mm is selected. From the rotation of the optimized response, the cylindrical cell is calculated based on the axisymmetric condition. This cell is radially replaced in the tire. The friction force between the tire and the road leads to bending in this cell. These cells act as a cantilever beam. During the movement and rotation of the tire, bending fatigue loading occurs in these cells. Therefore, the cell can be directly tested for fatigue. In the fatigue test, the sample is connected to the device with a screw at a distance of 1 cm from the beginning and the end. Therefore, this distance from the sample is assumed to be fixed (frozen area).

Since the length of the element is 25 mm, the dimensions of the plane in question were analyzed in a scaled manner. A total of 160 elements along the longitudinal axis and 18 elements along the transverse axis were selected (scale 100 times). Figure 11 shows the results of optimization in different constraints for this plane.

According to this figure, the sample with a residual weight limit of less than 20% had a discontinuity. The sample with a remaining weight of less than 40% cannot be manufactured with the FDM 3D printer due to the creation of very thin struts. Therefore, the sample with a 60% restriction was selected as the optimal sample.

The comparison of the results of the ABAQUS software and MATLAB code with the condition of remaining weight less than 60% is shown in Figure 12. This geometry is also optimized with multiple materials using the MATLAB code and the convergence results of its objective function are illustrated in Figure 13 and the response in a different step of the solution is depicted in Figure 14.

### 3.3. Entire Circumference of Tire

This code was also used to optimize the entire circumference of the tire. For this purpose, a square plane with dimensions of 650 × 650 mm was subjected to compressive and bending forces of 5 kN, similar to a real tire. According to the element size, 26 elements were selected in two directions. Figure 15 shows the results of tire optimization with one type of material. Due to this shape, the tire does not have continuity when weight limits of 20% and 40% are used for optimization process. Therefore, taking into account the lowest weight, the tire with 60% was selected as the optimal tire. The same objective function and constraint was used for the optimal distribution of PLA and TPU materials in the tire. The objective function convergence for the tire is shown in Figure 16 and the results are depicted in Figure 17.

Jang et al. [70] also performed topology optimization for tires with the objective function of static stiffness in terms of weight and volume constraints under pressure and tension, separately. The defined supports and tire dimensions were the same as in this study. However, since the optimization of the material distribution was not considered in the literature [70], the middle part of the tire, which included the tire core, was not considered. In other words, a hole was included in the tire (the inner rim) from the very beginning. Under an optimization formulation, several different patterns were obtained depending on the number of sections, volume fraction, and weighting factors. Among them, three representative patterns were chosen and analyzed for possible applications under working conditions. The value of the objective function in these cells was between 15 and 20 mm/N and the final answer was converged in the number of repetitions between 15 and 30. In contrast, in the current research, the value of the target function was 12 mm/N and the final response was calculated in the number of repetitions of 5.

This code was also used to optimize a rectangular plane [69]. After optimization, this plane became a semicircle. In addition to optimizing the distribution of two materials, they also implemented the optimization of the distribution of three materials with voids. In the optimization of two materials along with the void, the response was converged in the number of repetitions of 20. The reason for this could be the size of the element used. In this research, the element size was 8 mm. However, in the current research, the size of the element was 25 mm. This smaller element reduced the speed of convergence. In the code, a price value is also included, but the effect of this was not considered in the present study. In addition, 2D and 3D optimizations of a rectangular plane with 1 to 10 materials were implemented [71]. The objective function in these optimizations was the minimum compliance and volume constraint. This plane was subjected to various forces, including tensile and bending, in a 2D state.

In the 2D plane under bending load and two materials, the objective function converged in the number of repetitions of 10. This behavior is similar to the present work. Curved surfaces were also observed in the final response. The percentage of the final volume fraction was 35% to 60%, which is the same as the range of optimized percentages in the current research.

### 3.4. Outputs for The Quality of 3D Printing

The optimized cell of the square plane was fabricated in two sizes of 8 and 10 mm with a 3D FDM printer. Figure 18 shows the image obtained from the FE-SEM of the surface of these two cells. According to this figure, the mechanical adhesion between the PLA and TPU material is quite clear in both scales.

Holes were observed on the surface of the cells according to Figure 19. These holes are caused by two factors. The first is the high temperature during 3D printing, which causes the filament to evaporate and create bubbles on the surface of the sample. These bubbles cause defects [72]. The second factor is the manufacturing method. In 3D printing, parts are fabricated layer by layer. The improper connection of layers with each other causes holes in the sample. The holes caused by the manufacturing method were observed in connecting similar layers (TPU-TPU) and connecting different layers (PLA-TPU). In addition, the weak connection between layers in the FDM method was observed [72,73,74]. These holes can be minimized by using optimal print parameters, such as the layer height, the number of shells, and especially the extruder temperature [75].

According to Figure 20, the 3D printed cell with a size of 10 mm was of better quality. In the 3D printed cell with a size of 8 mm, the PLA layers were not well connected to the TPU layer. Various factors affect the adhesion of layers of different materials. Among others, these include print temperature, speed, and density [76,77]. However, these parameters are the same in both sizes. Therefore, it can be concluded that cell size also affects the adhesion of different materials in 3D printing. In other research, it was shown that the large-sized printed sample was of better quality [74].

Figure 21 shows the cell placement in the tire. In future research, the mechanical properties of this tire will be discussed. When force is applied to this non-pneumatic tire, the adjacent cell structures also have a strong effect on each other, which can affect the structural stress change. In further investigations, the mechanical properties of this tire including compressive and fatigue properties will be discussed. In addition, different arrangements in the unit cell will also be studied.

## 4. Conclusions

In this research, to design a non-pneumatic tire optimization was carried out for three types of geometries, including a square plane, a rectangular plane, and the entire circumference of the tire, and also using three materials: PLA, TPU, and void. The objective function in this optimization was minimum compliance and weight was a constraint. The results obtained were listed as follows:In the optimization of the square plane, the sample with the remaining weight constraint equal to 40% was selected as the optimal sample;In the optimization of the rectangular plane, the sample with the remaining weight constraint equal to 60% was selected as the optimal sample;In the optimization of the entire circumference of the tire, the sample with a remaining weight of less than 60% was selected as the optimal sample;The objective function for all three problems in the optimal distribution of materials was convergent and acceptable;PLA and TPU materials had complete mechanical adhesion, based on FE-SEM images. In future research, a pull-off test could be carried out to check this adhesion more accurately.

## Figures and Tables

**Figure 1 polymers-15-01927-f001:**
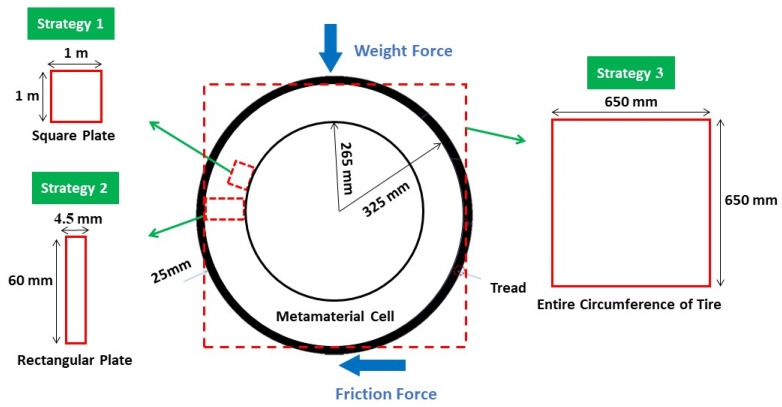
The dimensions of the tire plus the solution strategy.

**Figure 2 polymers-15-01927-f002:**
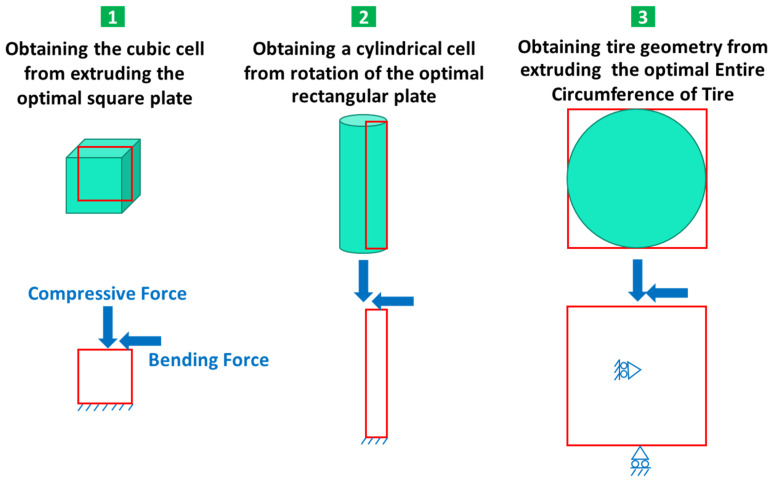
Problems defined in the MATLAB code.

**Figure 3 polymers-15-01927-f003:**
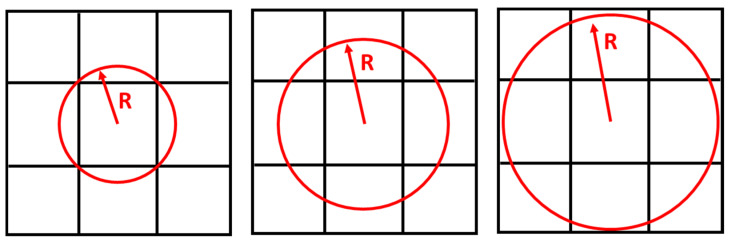
The radius filter to obtain a better objective.

**Figure 4 polymers-15-01927-f004:**
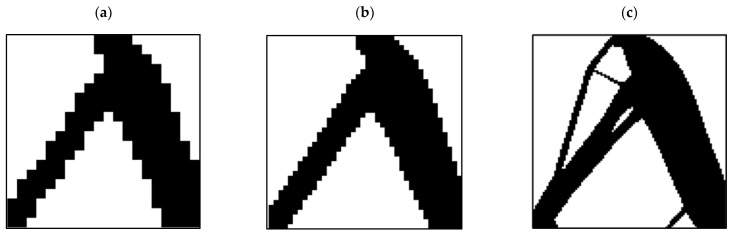
The effect of element size on the optimization response: (**a**) 50, (**b**) 25, and (**c**) 10 (mm).

**Figure 5 polymers-15-01927-f005:**
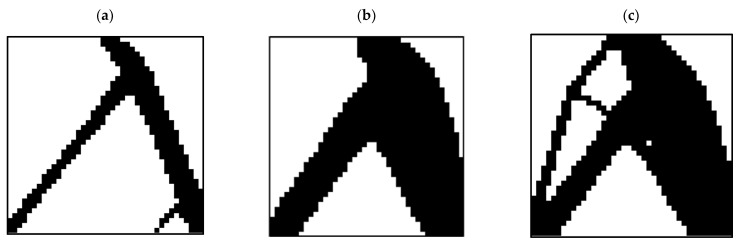
The response resulting from the optimization for the square plane in terms of (**a**) 20%, (**b**) 40%, and (**c**) 60% for the weight constraint.

**Figure 6 polymers-15-01927-f006:**
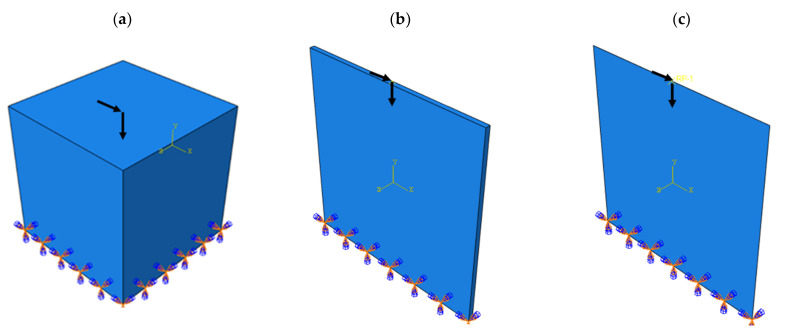
The problem defined in ABAQUS software: (**a**) 3D model, (**b**) 3D model with low thickness, and (**c**) 2D model.

**Figure 7 polymers-15-01927-f007:**
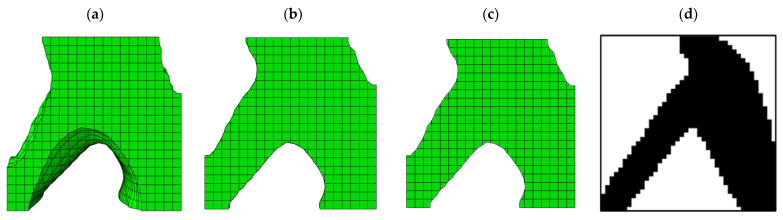
The comparison of the results obtained from ABAQUS software: (**a**) 3D model, (**b**) 3D model with low thickness, (**c**) 2D model, and (**d**) plus results of the MATLAB code.

**Figure 8 polymers-15-01927-f008:**
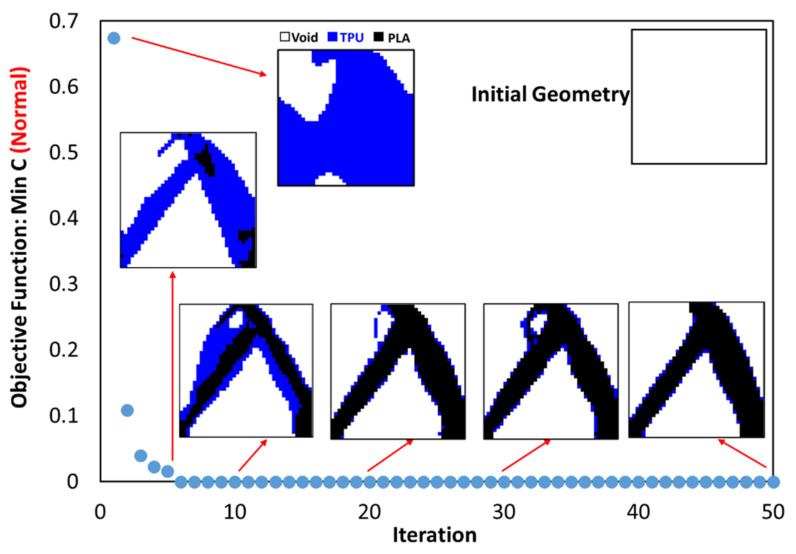
The convergence diagram of the objective function for optimal material distribution.

**Figure 9 polymers-15-01927-f009:**
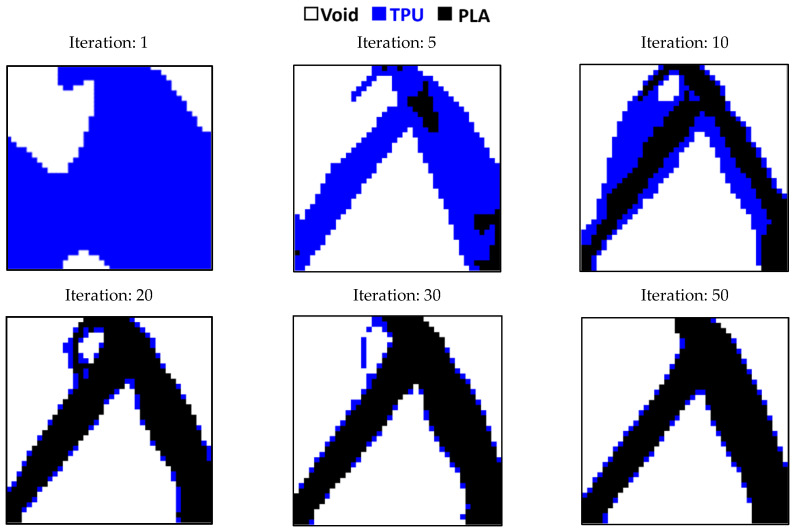
The Optimization of a square plane with multiple materials.

**Figure 10 polymers-15-01927-f010:**
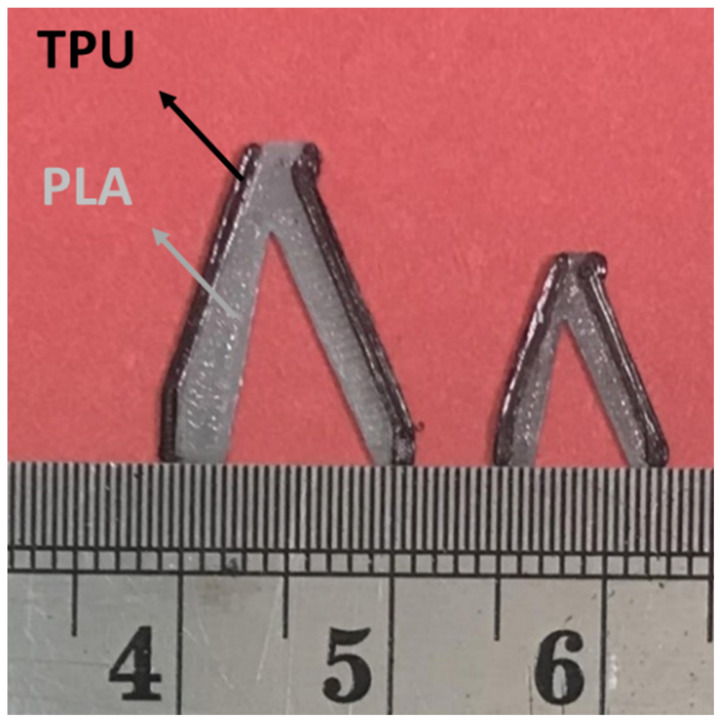
The bi-material metamaterial cells made by FDM 3D printer.

**Figure 11 polymers-15-01927-f011:**
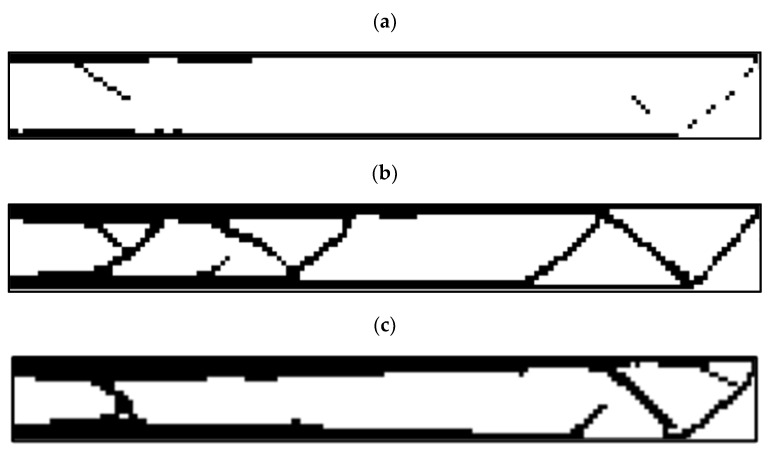
The results for the optimization of the rectangular plane in terms of (**a**) 20%, (**b**) 40%, and (**c**) 60% for the weight constraint.

**Figure 12 polymers-15-01927-f012:**
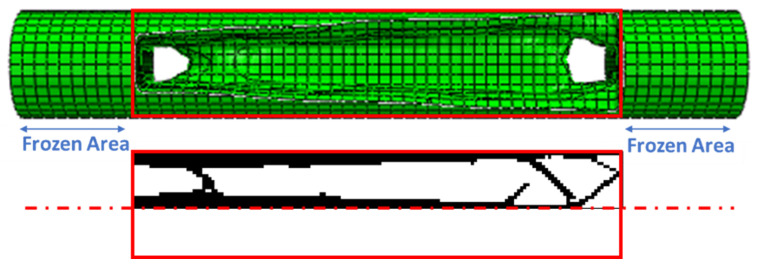
The comparison of the results obtained from ABAQUS software and MATLAB code for the rectangular plane. (Note: The red dotted line is the axisymmetric line for the sample).

**Figure 13 polymers-15-01927-f013:**
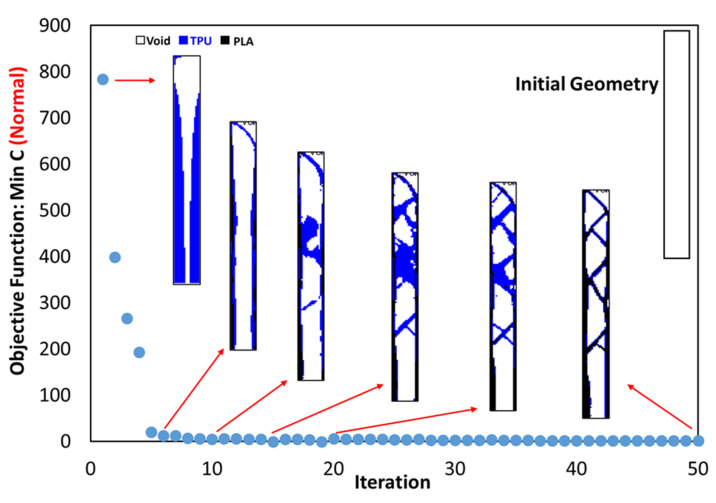
The convergence diagram of objective function for rectangular plane.

**Figure 14 polymers-15-01927-f014:**
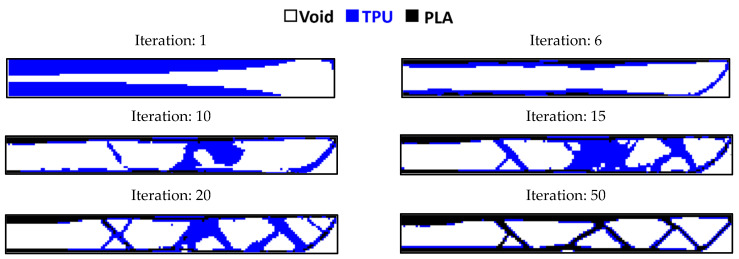
Optimizing the rectangular plane with multiple materials.

**Figure 15 polymers-15-01927-f015:**
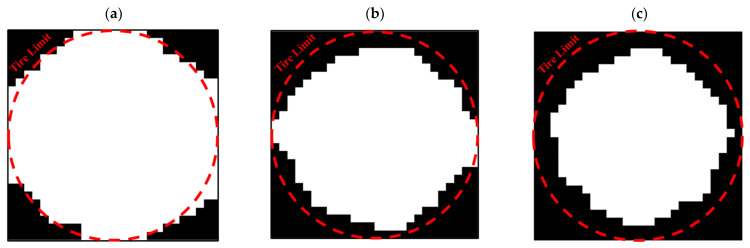
The optimized tire with one type of material: (**a**) 20%, (**b**) 40%, and (**c**) 60% of the weight constraint.

**Figure 16 polymers-15-01927-f016:**
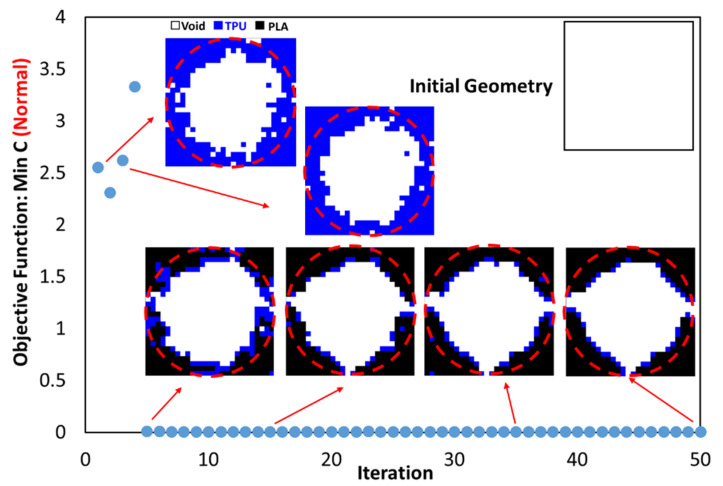
The convergence diagram of the objective function in tire optimization.

**Figure 17 polymers-15-01927-f017:**
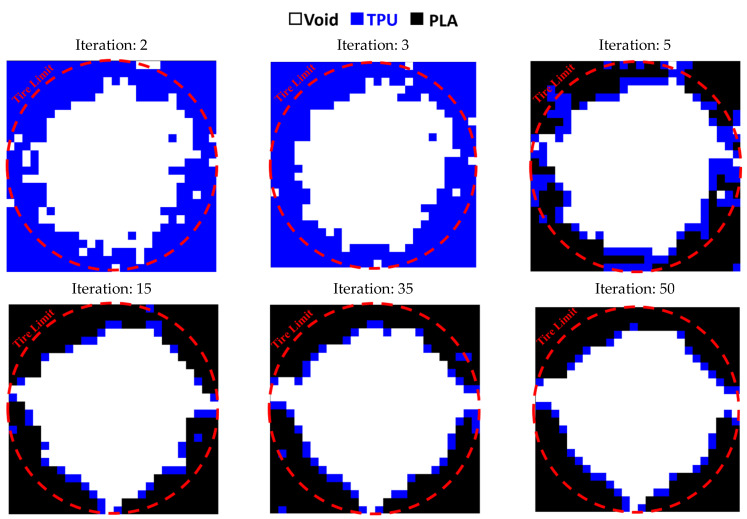
The results of the optimal distribution of materials in the tire.

**Figure 18 polymers-15-01927-f018:**
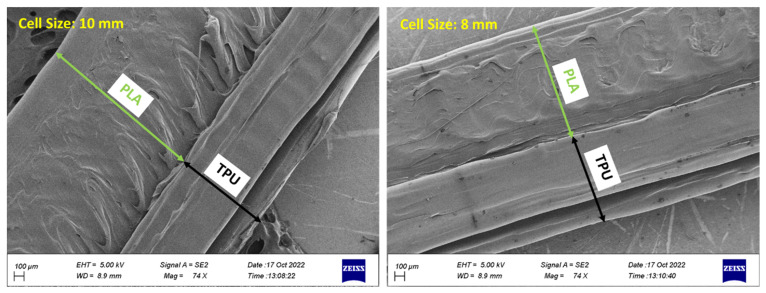
The FE-SEM image for connecting PLA/TPU materials.

**Figure 19 polymers-15-01927-f019:**
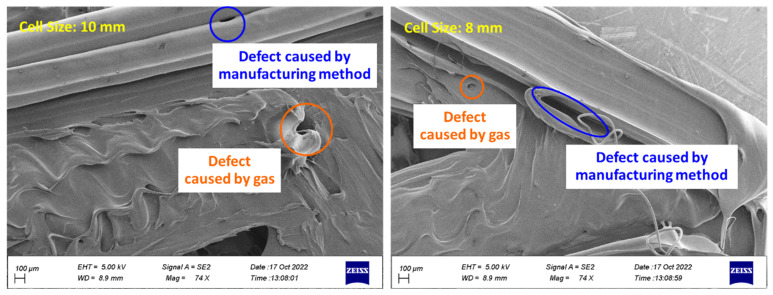
The defects fabricated on the surface of the sample during 3D printing.

**Figure 20 polymers-15-01927-f020:**
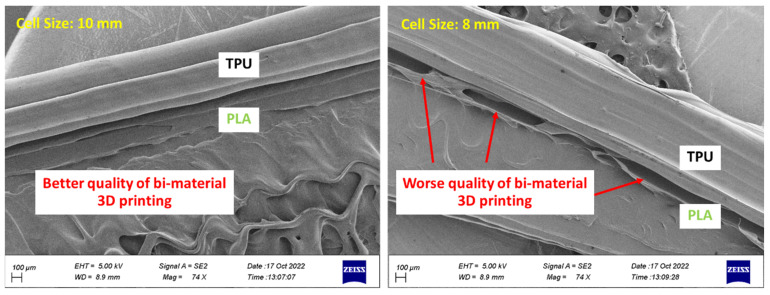
The effect of cell size on the adhesion of two PLA/TPU layers.

**Figure 21 polymers-15-01927-f021:**
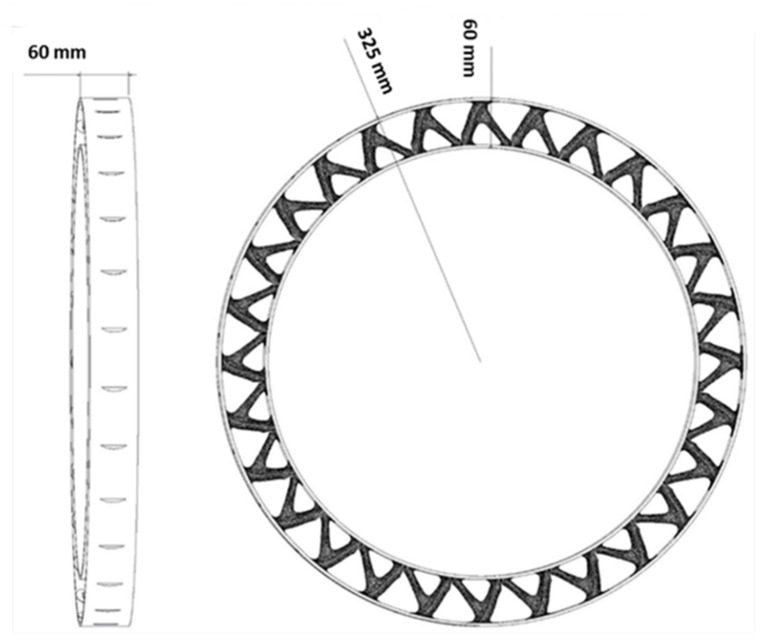
Arrangements of cells in the non-pneumatic tire.

**Table 1 polymers-15-01927-t001:** Parameters for 3D printing of PLA/TPU polymers.

Material	Speed (mm/s)	Nozzle Temperature (°C)	Infill (%)	Layer Height (mm)	Nozzle Diameter (mm)	Bed Temperature (°C)
PLA	20	180	100	0.2	0.2	25
TPU	20	220	100	0.2	0.2	25

**Table 2 polymers-15-01927-t002:** Mechanical properties of the materials studied.

Material	Yield Stress (MPa)	Elastic Modulus (MPa)
PLA	56.0 ± 2.5	3089.3 ± 100.7
TPU	4.1 ± 1.6	12.0 ± 0.6

**Table 3 polymers-15-01927-t003:** Optimization parameters in this research.

Objective	Constraint	R
Minimum Compliance	Remaining weight for 20, 40, and 60% of the total weight	1.2

## Data Availability

The data that support the findings of this study are available based on the request from the corresponding author. The experimental data are not publicly available due to restrictions and the privacy of research participants.

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
