# Peer review of "Multi-Material Metamaterial Topology Optimization to Minimize the Compliance and the Constraint of Weight: Application of Non-Pneumatic Tire Additive-Manufactured with PLA/TPU Polymers"

_polymers, 2023, doi:10.3390/polym15081927_

Round 1
Reviewer 1 Report
The present manuscript concerns a very interesting study on topological and materials optimization analysis for implementation in non-pneumatic tire technology.
The work is comprehensive and innovative.
I request minor revisions and suggestions prior to publication. Please see below:
- The abstract is too long and verbose it should be shortened according to the requirements of the journal (maximum 200 words).
- Can the authors specify the producers of the polymers used in the research and the characteristics of the 3D printer?
- Please pay attention to the data in Table 2. It seems that mean value and standard deviation of yield stress and Young modulus are reversed.
- Can the authors estimate the weight savings that would be achieved as a result of topological optimization?
- In the conclusions the authors mention that PLA and TPU are fully connected based on the SEM images. SEM is not a sufficient technique to prove the correct adhesion between the two polymers. On the "material" side, research should include mechanical adhesion or thermal analysis to establish compatibility between these components. This aspect could be considered as a next reserach step beyond those already mentioned by the authors in the text (which, however, should also be reported in the conclusions).
- In my opinion a noteworthy aspect regarding the research on non-pneumatic tires is the benefit in terms of environmental impact which is mentioned briefly on line 49. The possibility of engineering these systems with recyclable and reusable materials helps to reduce the serious environmental pollution problems due to the non-biodegradability of traditional tyres. Life cycle Assessment studies (please see https://doi.org/10.4271/2011-01-0093) can support this discussion. Furthermore, the discussion of the environmental impact perfectly matches with the new sustainable management and treatment actions for end-of-life tyres, such as the End of Waste decree in Europe and USA (please consider https://doi.org/10.3390/ma14247493) or the Waste Disposal and Public Cleansing Law in Japan (please consider https://doi.org/10.1007/s10163-011-0009-x). I suggest to authors to implement this information in the manuscript.
Author Response
Dear Editor-in-Chief,
The article, entitled “Multi-material Metamaterial Topology Optimization to Minimize the Compliance and the Constraint of Weight: Application of Non-pneumatic Tire Additive-manufactured with PLA/TPU Polymers”, is thankfully reviewed by your nice journal. First of all, thank you for giving us the chance of submitting a revision. Then also, we should thank the respected reviewers for their nice comments. All mentioned notes were addressed in the revised text and changes were also highlighted in yellow-colored sentences. Moreover, answers to all comments could be found in the following paragraphs.
Regards,
- Azadi, PhD.
Reviewer #1:
The present manuscript concerns a very interesting study on topological and materials optimization analysis for implementation in non-pneumatic tire technology.
The work is comprehensive and innovative.
I request minor revisions and suggestions prior to publication. Please see below:
Answer: First of all, the authors should thank the respected reviewer for his/her nice comments. We have tried our bests to address all comments in the revised article.
- The abstract is too long and verbose it should be shortened according to the requirements of the journal (maximum 200 words).
Answer: That issue is correct and therefore, the abstract is changed to the following one,
In non-pneumatic tires, metamaterial cells could replace the pneumatic part of the tire. In this research, to achieve a metamaterial cell suitable for a non-pneumatic tire with the objective function of increasing compressive strength and bending fatigue lifetime, an optimization was done for three types of geometries: square plane, rectangular plane, and the entire circumference of tire, besides three types of materials: polylactic acid (PLA), thermoplastic polyurethane (TPU), and void. The topology optimization was implemented by the MATLAB code in 2D mode. Finally, to check the quality of cell 3D printing and how they were connected, the optimal cell fabricated by the fused deposition modeling (FDM) method was evaluated using a field-emission scanning electron microscopy (FE-SEM). The results showed that in the optimization of the square plane, the sample with the minimum remaining weight constraint equal to 40% and in the optimization of the rectangular plane and the entire circumference of tire, the sample with the minimum remaining weight constraint equal to 60% was selected as the optimal case. From checking the quality of 3D printing of multi-materials, it was concluded that PLA and TPU materials were completely connected.
- Can the authors specify the producers of the polymers used in the research and the characteristics of the 3D printer?
Answer: According to the respected reviewer's opinion, it is added to the manuscript based on the following sentences,
The producer of both types of polymers was the YS company.
In the used 3D printer, the extruder is moved by the belt system in the X-Y axes and the bed is moved by the lead screw in the Z axis. This printer is able to print parts with dimensions of 550 x 600 x 700 mm. It has an accuracy of 50 and 40 μm in the X-Y and Z axes, respectively. Notably, this device was designed and manufactured by the research laboratory of Advanced Materials Behavior (AMB) at the Faculty of Mechanical Engineering, Semnan University, Semnan, Iran.
- Please pay attention to the data in Table 2. It seems that mean value and standard deviation of yield stress and young modulus are reversed.
Answer: Sorry! That issue is exactly correct and therefore, Table 2 is modified to the following one,
Table 2. Mechanical properties of the studied materials
|
Young modulus (MPa) |
Yield stress (MPa) |
Material |
|
3089.3±100.7 |
56.0±2.5 |
PLA |
|
12.0±0.6 |
4.1±1.6 |
TPU |
- Can the authors estimate the weight savings that would be achieved as a result of topological optimization?
Answer: With respect to the opinion by the reviewer, the optimization was done using the weight constraint. Therefore, the remaining weight is known and it is equal to the constraint used in the optimization. For example, in the optimization of the square plane, the sample with 40% constraint is selected. This means that the remaining weight is 40% of the total weight. However, in order to address the comment by the respected reviewer, the following text is added.
Notably, in Figure 7(a), the weight of the sample in 3D mode (ABAQUS) was 486 g. This is while the weight of the solid sample was 1240 g. Therefore, it can be concluded that the remaining weight was about 40% of the solid weight specimen.
- In the conclusions the authors mention that PLA and TPU are fully connected based on the SEM images. SEM is not a sufficient technique to prove the correct adhesion between the two polymers. On the "material" side, research should include mechanical adhesion or thermal analysis to establish compatibility between these components. This aspect could be considered as a next research step beyond those already mentioned by the authors in the text (which, however, should also be reported in the conclusions).
Answer: The respected reviewer is exactly correct. Therefore, with respected to the reviewer's opinion, Section 3.5 and conclusions are changed to the following parts.
According to this figure, the mechanical adhesion between PLA and TPU material is quite clear in both scales.
PLA and TPU materials had a complete mechanical adhesion, based on FE-SEM images. In the future research, the pull-off test could be done to check this adhesion, more accurate.
- In my opinion a noteworthy aspect regarding the research on non-pneumatic tires is the benefit in terms of environmental impact which is mentioned briefly on line 49. The possibility of engineering these systems with recyclable and reusable materials helps to reduce the serious environmental pollution problems due to the non-biodegradability of traditional tyres. Life cycle Assessment studies (please see https://doi.org/10.4271/2011-01-0093) can support this discussion. Furthermore, the discussion of the environmental impact perfectly matches with the new sustainable management and treatment actions for end-of-life tyres, such as the End of Waste decree in Europe and USA (please consider https://doi.org/10.3390/ma14247493) or the Waste Disposal and Public Cleansing Law in Japan (please consider https://doi.org/10.1007/s10163-011-0009-x). I suggest to authors to implement this information in the manuscript.
Answer: Thanks for the opinion by the respected reviewer. Then, it is added to the text, as follows,
Bras and Cobert [9] investigated the environmental impact of the Tweel tire during its life cycle, through the possibility of making these tires with recyclable and reusable materials. They concluded that non-pneumatic tires help to reduce the serious problems of the environmental pollution caused by the indestructibility of traditional tires [9]. Furthermore, the discussion of the environmental impact perfectly matches with the new sustainable management and treatment actions for end-of-life tires, such as the end of waste decree in Europe and USA [10] or the waste disposal and public cleansing law in Japan [11].

Reviewer 2 Report
Comments on the manuscript
This study focuses on optimizing the geometry and material of metamaterial cells for non-pneumatic tires to improve their compressive strength and bending fatigue life. Three types of geometry and materials were tested, with the optimal sample having a minimum remaining weight constraint of 40-60%. 3D printing was used to fabricate the cells, and both PLA and TPU materials were found to be completely connected.
Overall, the paper is scientifically intriguing and the experimental findings make contributions to the field of 3D functional materials. Additionally, the paper is technically sound, with well-supported conclusions and assertions. Therefore, the manuscript is within the scope of Polymers and I recommend publishing it, provided that the minor points are addressed.
1. Page 2. Quoting “Due to the complex structure of metamaterials, 3D printers are used to produce them.” This claim lacks reference support. Some recent advances in metamaterials fabricated by 3D printers can support your claim. Please see [Frenzel, Tobias, Muamer Kadic, and Martin Wegener. "Three-dimensional mechanical metamaterials with a twist." Science 358.6366 (2017): 1072-1074; Liang, Yao, et al. "Hybrid anisotropic plasmonic metasurfaces with multiple resonances of focused light beams." Nano Letters 21.20 (2021): 8917-8923]
2. Page 6. Section 3.1. Outputs for Square Plane. What challenges can arise in selective manufacturing methods, such as FDM, when creating optimized metamaterials with discontinuities, and how can these challenges be addressed? Please comment
3. Page 12. You mentioned the works of Jang et al. What are the main differences between the optimization approach used in this research and the one used by Jang et al. in their tire topology optimization study, and how do these differences affect the final optimization results?
4. Page 13. Section 3.5. Outputs for The Quality of 3D Printing. What are the causes of the holes observed on the surface of the cells fabricated with a 3D FDM printer, and how can they be minimized in future fabrication processes?
Author Response
Dear Editor-in-Chief,
The article, entitled “Multi-material Metamaterial Topology Optimization to Minimize the Compliance and the Constraint of Weight: Application of Non-pneumatic Tire Additive-manufactured with PLA/TPU Polymers”, is thankfully reviewed by your nice journal. First of all, thank you for giving us the chance of submitting a revision. Then also, we should thank the respected reviewers for their nice comments. All mentioned notes were addressed in the revised text and changes were also highlighted in yellow-colored sentences. Moreover, answers to all comments could be found in the following paragraphs.
Regards,
- Azadi, PhD.
Reviewer #2:
This study focuses on optimizing the geometry and material of metamaterial cells for non-pneumatic tires to improve their compressive strength and bending fatigue life. Three types of geometry and materials were tested, with the optimal sample having a minimum remaining weight constraint of 40-60%. 3D printing was used to fabricate the cells, and both PLA and TPU materials were found to be completely connected.
Overall, the paper is scientifically intriguing and the experimental findings make contributions to the field of 3D functional materials. Additionally, the paper is technically sound, with well-supported conclusions and assertions. Therefore, the manuscript is within the scope of Polymers and I recommend publishing it, provided that the minor points are addressed.
Answer: First of all, the authors should thank the respected reviewer for his/her nice comments. We have tried our bests to address all comments in the revised article.
- Page 2. Quoting “Due to the complex structure of metamaterials, 3D printers are used to produce them.” This claim lacks reference support. Some recent advances in metamaterials fabricated by 3D printers can support your claim. Please see [Frenzel, Tobias, Muamer Kadic, and Martin Wegener. "Three-dimensional mechanical metamaterials with a twist." Science 358.6366 (2017): 1072-1074; Liang, Yao, et al. "Hybrid anisotropic plasmonic metasurfaces with multiple resonances of focused light beams." Nano Letters 21.20 (2021): 8917-8923]
Answer: Thanks for the opinion by the respected reviewer. The first propose article is added in the text, as follows,
Metamaterials arise from the replication of cells next to each other [2, 5]. Due to the complex structure of metamaterials, 3D printers are used to produce them. For example, a type of metamaterial has been produced with a 3D printer that twist under pressure [5].
However, with respect, the second article was not added since it is about the magnetic metamaterials.
- Page 6. Section 3.1. Outputs for Square Plane. What challenges can arise in selective manufacturing methods, such as FDM, when creating optimized metamaterials with discontinuities, and how can these challenges be addressed? Please comment
Answer: With respect to the reviewer's opinion, the possible challenges for producing discontinuous structures with the FDM method are stated in section 3.1, according to the following sentences,
According to this figure, the sample with less remaining weight equal to 20% and 60% has discontinuity. This discontinuity requires support according to the selective manufacturing method (FDM), which in metamaterials is very difficult, if not impossible to separate these types of supports from the main part due to the repetition of the cell and the complexity of the part.
However, the solution method of these challenges was not stated before, which is now added to the text, as follows,
One of the ways to overcome this challenge is to use the constraint related to additive manufacturing. In these types of constraints, the overhang angle (the angle between the material and the horizon line) is defined. In this case, angles greater than 45° that require support will not be fabricated along with the specimen.
- Page 12. You mentioned the works of Jang et al. What are the main differences between the optimization approach used in this research and the one used by Jang et al. in their tire topology optimization study, and how do these differences affect the final optimization results?
Answer: With respected to the reviewer's opinion, this text is changed to the following sentences,
Jang et al. [70] also performed the topology optimization for tires with the objective function of static stiffness in terms of weight and volume constraints under pressure and tension, separately. The defined supports and tire dimensions were the same to this study. However, since the optimization of the material distribution was not considered in the literature [70], the middle part of the tire, which included the tire core, was not considered. In other words, a hole was included in the tire (the inner rim) from the very beginning stage. Under an optimization formulation, several different patterns were obtained depending on the number of sections, volume fraction, and weighting factors. Among them, three representative patterns were chosen and analyzed for possible applications under working conditions. The value of the objective function in these cells was between 15 and 20 mm/N and the final answer was converged in the number of repetitions between 15 and 30. While in the current research, the value of the target function was 12 mm/N and the final response was calculated in the number of repetitions of 5.
- Page 13. Section 3.5. Outputs for The Quality of 3D Printing. What are the causes of the holes observed on the surface of the cells fabricated with a 3D FDM printer, and how can they be minimized in future fabrication processes?
Answer: Thanks for the opinion by the respected reviewer. The reasons for creating the holes were written before, as follows,
These holes are caused by two factors. The first factor is caused by the high temperature during 3D printing, which causes the filament to evaporate and create bubbles on the surface of the sample. These bubbles cause defect [72]. The second factor is caused by the manufacturing method.
However, the factors of reducing these holes were not stated, which are added in the revised article, as follows,
These holes can be minimized by using optimal print parameters, like the layer height, the number of shells, and especially the extruder temperature [75].
